# Effect of Geumgwe-Sinkihwan on Renal Dysfunction in Ischemia/Reperfusion-Induced Acute Renal Failure Mice

**DOI:** 10.3390/nu13113859

**Published:** 2021-10-28

**Authors:** Byung Hyuk Han, Hyeon Kyoung Lee, Se Hoon Jang, Ai Lin Tai, Youn Jae Jang, Jung Joo Yoon, Hye Yoom Kim, Ho Sub Lee, Yun Jung Lee, Dae Gill Kang

**Affiliations:** 1Hanbang Cardio-Renal Syndrome Research Center, Wonkwang University, 460 Iksan-daero, Iksan 54538, Jeonbuk, Korea; arum0924@naver.com (B.H.H.); gorud0170@naver.com (H.K.L.); wkdtpgns321@naver.com (S.H.J.); doflal0324@naver.com (A.L.T.); j8626@naver.com (Y.J.J.); mora16@naver.com (J.J.Y.); hyeyoomc@naver.com (H.Y.K.); host@wku.ac.kr (H.S.L.); 2College of Oriental Medicine and Professional Graduate School of Oriental Medicine, Wonkwang University, 460 Iksan-daero, Iksan 54538, Jeonbuk, Korea

**Keywords:** geumgwe-sinkihwan, ischemia/reperfusion, creatinine, NGAL, KIM-1, fibrosis, cytokine, inflammasome

## Abstract

Renal ischemia-reperfusion (I/R) injury is an important cause of acute renal failure (ARF). Geumgwe-sinkihwan (GSH) was recorded in a traditional Chines medical book named “Bangyakhappyeon” in 1884. GSH has been used for treatment for patients with diabetes and glomerulonephritis caused by deficiency of kidney yang and insufficiency of kidney gi. Here we investigate the effects of GSH in mice model of ischemic acute kidney injury. The mice groups are as follows; sham group: C57BL6 male mice, I/R group: C57BL6 male mice with I/R surgery, GSH low group: I/R + 100 mg/kg/day GSH, and GSH high group: I/R + 300 mg/kg/day GSH. Ischemia was induced by clamping both renal arteries and reperfusion. Mice were orally given GSH (100 and 300 mg/kg/day) during 3 days after surgery. Treatment with GSH significantly ameliorated creatinine clearance, creatinine, and blood urea nitrogen levels. Treatment with GSH reduced neutrophil gelatinase associated lipocalin (NGAL) and kidney injury molecule-1 (KIM-1), specific renal injury markers. GSH also reduced the periodic acid–Schiff and picro sirius red staining intensity in kidney of I/R group. Western blot and real-time RT-qPCR analysis demonstrated that GSH decreased protein and mRNA expression levels of the inflammatory cytokines in I/R-induced ARF mice. Moreover, GSH inhibited protein and mRNA expression of inflammasome-related protein including NLRP3 (NOD-like receptor pyrin domain-containing protein 3, cryoprin), ASC (Apoptosis-associated speck-like protein containing a CARD), and caspase-1. These findings provided evidence that GSH ameliorates renal injury including metabolic dysfunction and inflammation via the inhibition of NLRP3-dependent inflammasome in I/R-induced ARF mice.

## 1. Introduction

The kidney is very important for the control of body-fluid homeostasis. The dysregulation of its function causes pathological conditions such as hypertension, nephrogenic diabetes insipidus, nephrotic syndrome, and acute renal failure (ARF) [1,2,3,4]. Typically, ARF, defined as acute degradation of kidney function, is due to acute tubular necrosis, which results in kidney failure due to ischemic or nephrotoxic abuses. Acute kidney injury (AKI) is a major source of ARF, and is a general clinical syndrome with significant morbidity, mortality, and medical costs [5,6,7]. AKI also can be a risk of chronic kidney disease and end-stage renal disease [8]. However, there is no specific therapeutic strategy for AKI, therefore, it is necessary to search for new and effective drugs for AKI patients. Ischemia/reperfusion injury is the main cause of AKI, and ARF caused by renal ischemia/reperfusion is commonly defined by a severe decrease in the glomerular filtration rate (GFR), renal blood flow, and a down-regulating in the urinary concentrating ability [9,10,11,12].

Pathological consequences of ARF result in reduced glomerular filtration [13]. After ischemia/reperfusion injury, the cell damage mechanism causes the release of inflammatory mediators and an increase in oxidative stress through the interaction between tubule and blood vessels [14]. Clinically, elevated serum creatinine and decreased urinary tract are major markers of ARF. When ARF occurs, kidney injury molecule-1 (KIM-1) is a typical renal injury marker for increased expression in blood and urine excretion [15]. Neutrophil gelatinase-associated lipocalin (NGAL) is also increased in kidney damage such as infections, malignant tumors, and neuropathic tubules [16,17]. NGAL is a powerful marker for early treatment of ARF and for improving prevention [18,19].

Additionally, ischemic/reperfusion-induced kidney diseases develop fibrosis and lesions, which lead to kidney dysfunction and inflammation [20]. NOD-like receptor family pyrin domain containing 3 (NLRP3) has been linked to the pathogenesis of various inflammatory disorders when activated [21]. The NLRP3-dependent inflammasome is a tripartite complex which consists of NLRP3, apoptosis--associated speck--like protein containing a CARD domain (ASC), and procaspase--1 [22,23,24]. The NLRP3 inflammasome is a family of the cytoplasmic pattern recognition receptor that plays an important role of cytokines production [21] Therefore, NLRP3 inflammasome and proinflammatory cytokines directly affect renal tubular epithelium and cause renal dysfunction.

Recently, various chemical drugs and compounds have been used to treat acute and chronic renal failure. However, those therapeutic treatments have various side effects. Therefore, interest in treatment using natural products without side effects is increasing, and for this purpose, this study was conducted through traditional medicine using natural herbs. Sinkihwan-gamibang (腎氣丸加味方, SKHGMB) consists of three prescriptions; Sinkihwan (腎氣丸, SKH), Geumgwe-sinkihwan (金櫃腎氣丸, GSH), and Jesaeng-sinkihwan (濟生腎氣丸, JSH). SKHGMB has been described in a traditional Korean medical book named “Bangyakhappyeon” (方藥合編, 1884). According to records, GSH is composed of eight dried natural herbs; *Rehmannia glutinosa* (rhizome), *Discorea batatas* (rhizome), *Cornus officinalis* (fruit), *Poria cocos* (Schw.) *Wolf* (Hoelen), *Alisma orientale* (rhizome), *Paeomia suffructicosa* (cortex), *Achyranthes aspera* (root), and *Plantago ovata* (root); in a ratio of 8:4:4:3:3:3:1:1. SKHGMB has been used for the treatment of chronic nephritis, such as pollakiuria, generalized edema, and decreased kidney function. JSH ameliorated tubular injury including renal dysfunction in ischemia/reperfusion-induced ARF mice [25]. GSH has been used for the treatment of patients with diabetes and glomerulonephritis caused by deficiency of kidney yang (陽) and insufficiency of kidney gi (氣). However, the pharmacological effect of GSH has not been studied. Therefore, the present study was designed to investigate the protective role of GSH against ARF.

## 2. Materials and Methods

### 2.1. Preparation of Geumgwe-Sinkihwan

GSH is composed of 8 herbal medicines (Table 1). The herbs were purchased from the Chung Woong Cooperative Association (Imsil, Korea) and specimen voucher number was HBA 192-10. The prescription was boiled with 3 L of distilled water. After centrifugation, supernatant was concentrated by evaporator (N-11, Tokyo Rikakikai, Tokyo, Japan). Extract (101.59 g) was lyophilized with a freeze-drier (PVTFD10RS, IlShinBioBase, Yangju, Korea). The yield rate was 8.192% and GSH was kept at 4 °C.

### 2.2. Experimental Animals

All animal experimental protocols were approved by the Institutional Animal Care and Utilization Committee for Medical Science of Wonkwang University (WKU16-59). 9-week-old male C57BL6 (C57 black 6) mice were purchased from Koatech (Pyeongteak, Korea) and then housed with automatic temperature, lighting (12 h light/dark cycle), and humidity (22 ± 2 °C, 50–60%). After 3 days acclimatization, the animals were randomly divided as follows (*n* = 9): (1) sham (normal), (2) ischemia/reperfusion (ARF), (3) GSH low (I/R + GSH 100 mg/kg/day), (4) GSH high (I/R + GSH 300 mg/kg/day). To induce ischemia/reperfusion ARF, both renal arteries were clamped for 25 min under isoflurane anesthesia (Harvard Apparatus, Small Animal Ventilator, MA, USA). Sham mice underwent operation without clamping the renal arteries. Each groups of animals were orally administered with distilled water or GSH, respectively, for 3 days. During the experiments, all mice were housed separately in a metabolic cage for quantitative urine collections. All mice were sacrificed after 12 h fasting, and blood samples were collected from facial vein.

### 2.3. Measurement of Plasma and Urine Biochemical Parameters

Plasma levels of blood urea nitrogen (BUN), plasma creatinine (CRE), and urine creatinine (Ucr) were analyzed using automated chemistry analyzer (NX700, FUJIFILM, Tokyo, Japan). The values of creatinine clearance (Ccr) were confirmed for detection of renal function and calculated as follows (urine volume (UV), creatinine (Cr), body weight (B.W)).
$$\frac{\mathrm{UV}\;\left(\mathrm{mL}\right)*\mathrm{Ucr}\;\left(\mathrm{mg}/\mathrm{mL}\right)}{\mathrm{Cr}\;\left(\mathrm{mg}/\mathrm{mL}\right)*B.W\;\left(\mathrm{kg}\right)*\min}$$

### 2.4. Morphological Measurement of Renal Tissues

Kidneys were fixed with 10% formalin (Junsei Chemical, Tokyo, Japan), and embedded in paraffin. Cutting 5 μm slides were stained with periodic acid–Schiff (PAS) reagent (Sigma, MO, USA) and Picrosirius Red Stain Kit (PolySciences, PA, USA), respectively. All slide samples were reviewed and scored by the unaware pathologist. Histological changes and fibrosis were analyzed by using the EVOS M5000 imaging system (Thermo Fisher Scientific, Waltham, WA, USA). Areas of glomerular injury in each image was quantified using ImageJ software (NIH, Bethesda, MD, USA).

### 2.5. Western Blot Analysis

Kidney tissue lysates (40 μg) were run on 10% SDS-polyacrylamide gel electrophoresis, subsequently transferred to nitrocellulose membrane. Blots were blocked with 5% BSA and then exposed to the appropriate primary antibody in dilutions suggested from the commercial supplier. Primary antibody KIM-1, NGAL, TNF-α, IL-1β, IL-6, cryopyrin (NLRP3), ASC, pro-caspase-1, and β-actin were purchased from Santa Cruz Biotechnology (Dallas, TX, USA). Primary antibodies were probed with horseradish peroxidase conjugated goat anti-rabbit-IgG or anti-mouse-IgG. Visualization was performed by chemiluminescence (EzWestLumi plus, ATTO, NY, USA). Capturing image was achieved by gel image system (iBright FL100, Thermo Fisher Scientific, Waltham, MA, USA).

### 2.6. Quantitative Real-Time Reverse Transcription-PCR (Real-Time RT-qPCR)

Isolation of total mRNA was conducted using a commercially available kit. cDNA synthesis was performed by reverse transcription at 37 °C for 60 min, 94 °C for 5 min, respectively. The sequences of primers and probes were as follows: TNF-α (forward: 5′-GAC AAG CCT GTA GCC CAC GT-3′, reverse: 5′-ACA AGG TAC AAC CCA TCG GC-3′), IL-1β (forward: 5′-TGA CGG ACC CCA AAA GAT GA-3′, reverse: 5′-ACA GCT TCT CCA CAG CCA CA-3′), IL-6 (forward: 5′-GAG GAT ACC ACT CCC AAC AGA CC-3′, reverse: 5′-AAG TGC ATC ATC GTT GTT CAT ACA-3′), NLRP3 (forward: 5′-AGG AGA AAG AAG AAG AGA GGA-3′, reverse: 5′-AGA GAC CAC GGC AGA AGC-3′), ASC (forward: 5′-CAG CCA TCC CGT GCC TCC AGA TCA C -3′, reverse: 5′- CCA GAG AAA TGG AGT GGG CAT CAA G-3′), caspase-1 (forward: 5′-CGT CTT GCC CTC ATT ATC TG-3′, reverse: 5′-TCA CCT CTT TCA CCA TCT CC-3′), and GAPDH (forward: 5′-CGA GAA TGG GAA GCT TGT CAT C-3′, reverse: 5′-CGG CCT CAC CCC ATT TG-3′). The real-time RT-qPCR was carried out in a mixture of template cDNA and 50 nM primers, and SYBR Green PCR pre-mix according to the manufacturer instruction (Intron, Seongnam, Korea). The amplification steps were complied with 45 cycles at 94 °C, 20 s; 60 °C, 20 s; 72 °C, 30 s by using the Step-One™ Real-Time PCR System (Applied Biosystems, CA, USA). Each sample was measured in triplicate and resulting data was normalized against GAPDH mRNA abundance.

### 2.7. Immunohistochemical Analysis of Renal Tissues

Kidney were fixed with 10% formalin and embedded in paraffin. Slides were immersed in 3% hydrogen peroxide for 10 min to suppress endogenous peroxidase activity. Sections were stained with mouse- and rabbit-specific HRP/DAB (APC) Detection IHC kit (Abcam, MA, USA). Staining was conducted by manufacturer protocol. Slides were visualized by using EVOS M5000 imaging system.

### 2.8. Statistical Analysis

The results were displayed as mean ± standard error (S.E.) and were analyzed using one-way ANOVA, Dunnett’s test, or Student’s *t*-test using version 10 Sigma Plot software to determine any significant differences. *p* < 0.05 was represented as statistical significance.

## 3. Results

### 3.1. Effect of GSH on Renal Function Parameters

To evaluate the beneficial effects of GSH on renal function markers, BUN and creatinine in plasma were measured. As shown in Table 2, compared with the sham group, the I/R group displayed increased plasma BUN and creatinine levels. However, these levels were significantly decreased by the oral administration of GSH. In addition, creatinine clearance (CCr) levels, which are an indicator of renal function, were increased by GSH as opposed to the I/R group (Figure 1A). As shown in Figure 1B, kidney weight/body weight (KW/BW) and kidney sizes were significantly increased in I/R-induced ARF group as compared with the sham group. However, GSH significantly decreased KW/BW and kidney size. Urine volume significantly decreased in the I/R-induced ARF group. However, these levels were significantly increased by GSH treatment (Figure 1C). Finally, urine creatinine levels also were decreased by GSH treatment (Figure 1D).

### 3.2. Effect of GSH on Renal Morphology and Renal Fibrosis

To investigate the effect of GSH on the morphology of kidney, histological changes were measured by staining with PAS staining. As shown in Figure 2, morphological damage to the extracellular matrix (ECM), including basement membrane and mesangial matrix, which were not observed in sham group, has been identified in the I/R group. In the GSH treatment group, as in the sham group, some damage to the extracellular matrix, including the mesenchymal substrate, was observed, but this was significantly lower than in the I/R group. In addition, to measure the fibrosis on the kidney, kidneys were stained using picrosirius red (Figure 3). As a result, color calm increased in the kidney in the I/R group, and improvement effects could be observed in GSH-treated mice. These results suggest that treatment with GSH attenuates renal histological changes and renal fibrosis.

### 3.3. Effect of GSH on I/R-Induced Renal Nephropathy

To determine whether GSH improve renal nephropathy, protein expressions of KIM-1 and NGAL, markers of critical nephropathy, were examined by using Western blot analysis (Figure 4A). The result showed that I/R-induced ARF group exhibited increased levels of KIM-1 and NGAL. However, treatment with GSH suppressed these increases. Similarly, mRNA expression levels of KIM-1 and NGAL were significantly increased in the I/R-induced ARF group as compared with sham group. However, GSH significantly decreased mRNA levels (Figure 4B). The I/R group exhibited significantly increased expressions of KIM-1 and NGAL in immunohistochemistry compared with the sham group (Figure 4C). However, treatment with GSH significantly attenuated the expression of KIM-1 and NGAL.

### 3.4. Effect of GSH on Renal Inflammatory Cytokines

I/R induces the activation of inflammatory cytokines in the kidney. In this present study, the regulation of renal inflammatory markers such as TNF-α, IL-1β, and IL-6 were investigated by Western blot and real-time qPCR. The I/R group showed a significant increase in these markers compared with the sham (normal) group. However, administration of GSH resulted in the decreased protein expression of these cytokines (Figure 5A). GSH also decreased the mRNA expression of these inflammatory markers (Figure 5B). This result suggests that renal inflammation was involved in various cytokine productions when ischemia occur, and GSH may ameliorate renal inflammation by direct inhibition of the expressions of inflammatory markers.

### 3.5. Effect of GSH on Regulation of NLRP3 Inflammasome

The NLRP3-dependent inflammasome plays a key role in renal inflammation. Western blot analysis was performed to confirm that the effect of GSH on the protein expressions of the NLRP3 inflammasome (Figure 6A). Among the NLRP3 inflammasome signaling factors, NLRP3, ASC, and caspase-1 protein expression were activated in the I/R-induced ARF group. However, these protein expressions were inhibited by GSH. Similarly, compared with the sham group, the NLRP3 inflammasome mRNA expressions were increased in the I/R group (Figure 6B). However, these mRNA expressions were significantly decreased by oral administration of GSH.

## 4. Discussion

Ischemia/reperfusion injury is one of the most common sources of acute renal failure and acute kidney injury, potentially fatal conditions [26]. Until now, the only possible therapeutic strategies were kidney transplantation and dialysis. Thus, there is an increasing need to find an effective drug for the improvement of IRI-induced AKI. A previous study reported that GSH has an anti-cancer effectiveness and can be clinically a potent antimetastatic drug for the prevention and treatment of cancer [27]. However, the effect of GSH on renal metabolic dysfunction and injury has not been reported yet. Therefore, this study was performed to clarify whether GSH has a protective effect in mice with I/R-induced AFR and improves renal function.

AKI is diagnosing based on blood urea nitrogen (BUN) and serum creatinine levels [28]. Creatinine clearance (CCr) also is one of most the important marker for AKI [29]. The hypertrophy of the kidney and the increase in the ratio of kidney to body weight are symptoms in the early stages of acute renal failure [30]. In this study, GSH was administered in I/R-induced AKI mice and we found that GSH ameliorated renal function markers such as Cr and BUN. The treatment of GSH also recovered the levels of CCr and KW/BW. These results suggest that GSH is effective in the improvement of AKI through the regulation of renal functional biomarkers by suppressing the filtration and re-absorption function of the kidneys.

Acute kidney injury can lead to glomerular mesangium expansion, resulting from hypertrophy and hyperplasia of glomerular mesangial cells and epithelial cells, and accumulation of extracellular matrix [31]. Therefore, this study identified kidney injury and the effect of recovering kidney damage by GSH through PAS staining. As a result, damage to glomerular was confirmed in I/R-induced AKI group. Furthermore, observation of inner medulla and outer medulla confirm the morphological change and the formation of vacuoles. GSH has the effect of reducing glomerular damage and the morphological damage of kidneys. In addition, renal fibrosis is when tissue is damaged and thickened by inflammation of tissued caused by external or internal stimuli [32]. Fibrosis is also induced by cytokines, which increase the accumulation and collagen synthesis of extracellular substrates, leading to sclerosis [33]. In this study, the level of fibrosis was observed thorough picrosirius red staining to confirm the inhibitory effect of renal fibrosis by GSH. This study demonstrated that treatment with GSH reduced the level of fibrosis in I/R-induced mice.

Typical indicators of kidney damage and renal dysfunction include neutrophil gelatinase-associated lipocalin (NGAL) and kidney injury molecule-1 (KIM-1). The levels of NGAL and KIM-1 increase very quickly after rapid changes in renal function [34]. NGAL and KIM-1 are biological molecules that can indicate the exact extent of kidney damage caused by acute renal failure [35,36], and NGAL especially is a gene that increases due to acute kidney damage [37]. KIM-1 is also a protein that indicates the degree of renal proximal injury [38]. Therefore, NGAL and KIM-1 are used as very accurate evaluation metrics related to acute renal failure. In this study, the effects of GSH on the expression of NGAL and KIM-1 were analyzed. In I/R-induced AKI mice group, the expression of NGAL and KIM-1 was found to increase and decrease significantly by treating with GSH. These results indicate that GSH is effective in improving kidney damage.

A previous study reported that inflammation plays a role in the development of IRI-induced AKI [39]. Cytokines have been reported as key mediators of inflammation in various tissue injuries. In the present study, levels of cytokines such as TNF-α, IL-1β, and IL-6 were significantly elevated in renal tissues following I/R, however, GSH significantly suppressed the expression levels of cytokines. In addition, NLRP3-dependent inflammasomes play a critical role in acute or chronic diseases, therefore, interference of NLRP3 inflammasome signaling could regulate the renal injury [40]. In the present study, Western blot and real-time RT-qPCR analysis revealed that GSH reduced I/R-induced NLRP3-dependent inflammasome. This result demonstrated that GSH significantly inhibited I/R-induced inflammasome-related proteins including NLRP3, ASC, and caspase-1. Therefore, this suggests that GSH decreases renal inflammatory response by reducing NLRP3-dependent inflammasome complex formation after ischemic injury.

## 5. Conclusions

The results of this study show the effect of improving kidney damage and kidney function by treating with GSH, which is regarded to be effective as an inhibitor of ischemic diseases by suppressing kidney damage in animal models with acute renal failure due to I/R. Therefore, further research on this may lead to the development of GSH as a good supplement to the treatment of ischemic renal disease.

## Figures and Tables

**Figure 1 nutrients-13-03859-f001:**
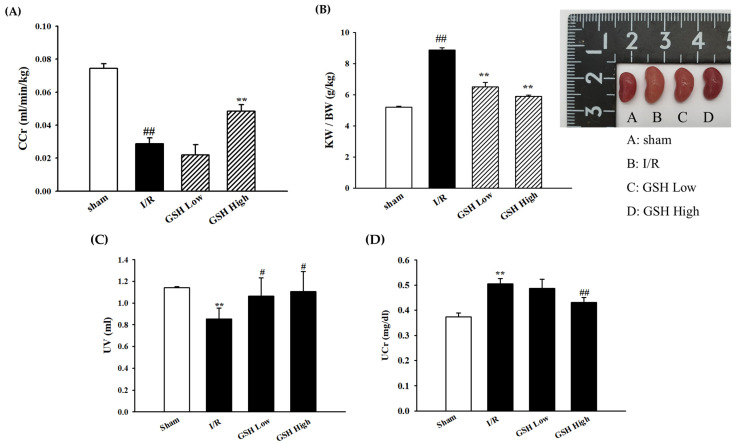
Effect of GSH on creatinine clearance (CCr) (**A**), KW/BW (**B**), UV (**C**), and urine creatinine (UCr) (**D**) in mice. Values are expressed as mean ± S.E. (*n* = 9 per group). ** *p* < 0.01 vs. sham group; ^#^
*p* < 0.05, ^##^
*p* < 0.01 vs. I/R group.

**Figure 2 nutrients-13-03859-f002:**
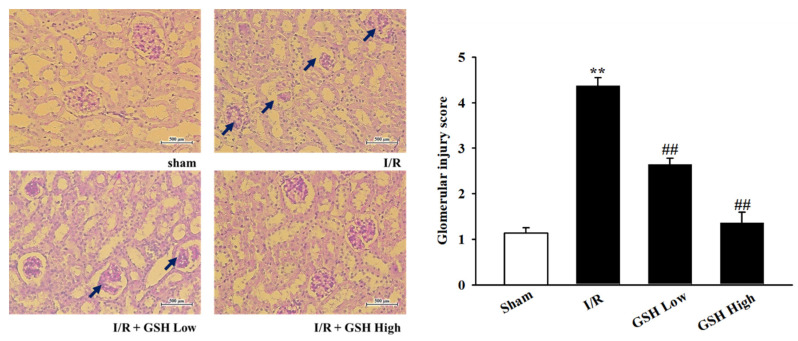
Effect of GSH on renal injury. Representative photomicrographs showed PAS-stained tissues (magnification 200×, scale bar: 500 μm). Right bar graph indicates the quantitative assessment of glomerular injury. ^##^ *p* < 0.01 vs. sham group; ** *p* < 0.01 vs. I/R group.

**Figure 3 nutrients-13-03859-f003:**
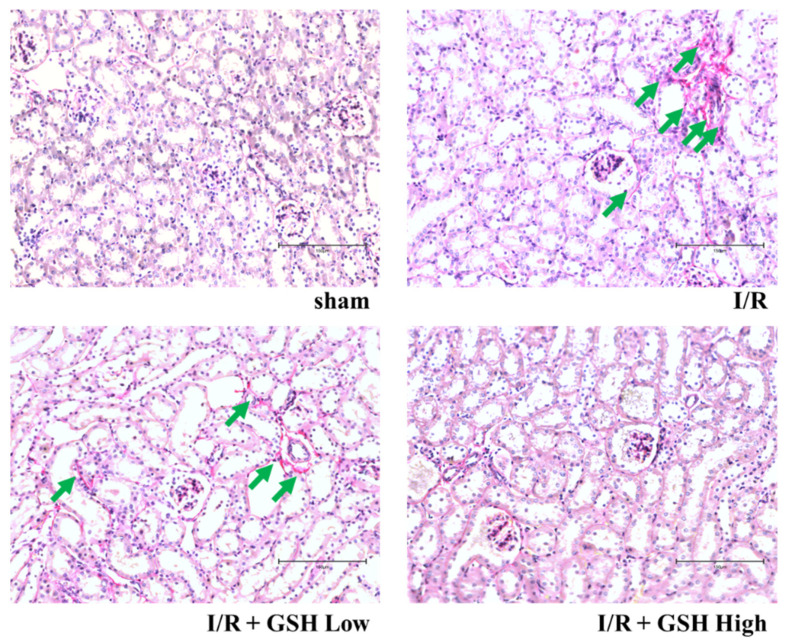
Effect of GSH on renal fibrosis in I/R-induced renal failure. Representative images of Picrosirius red staining for collagen deposition analysis in I/R-induced renal failure mice. Cardiomyocytes (purple) and collagen fibers (green arrow) were shown (magnification 400×, scale bar: 150 μm).

**Figure 4 nutrients-13-03859-f004:**
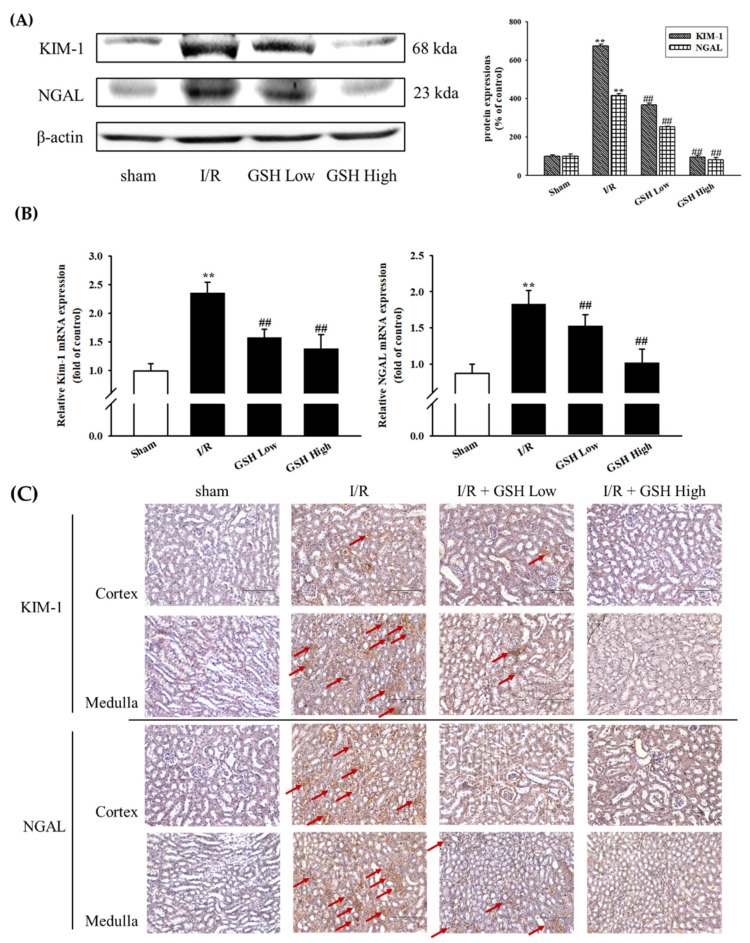
Effect of GSH on I/R-induced renal nephropathy. Protein levels of nephropathy biomarkers such as KIM-1 and NGAL were evaluated by Western blot analysis (**A**). Relative KIM-1 and NGAL mRNA expressions were analyzed by real-time RT-qPCR (**B**). The bottom panel show immunohistochemistry staining (magnification 400×, scale bar: 150 μM).KIM-1 and NGAL expressions were shown (red arrow) (**C**). Values were expressed as mean ± SE (*n* = 3). ^##^ *p* < 0.01 vs. sham group; ** *p* < 0.01 vs. I/R group.

**Figure 5 nutrients-13-03859-f005:**
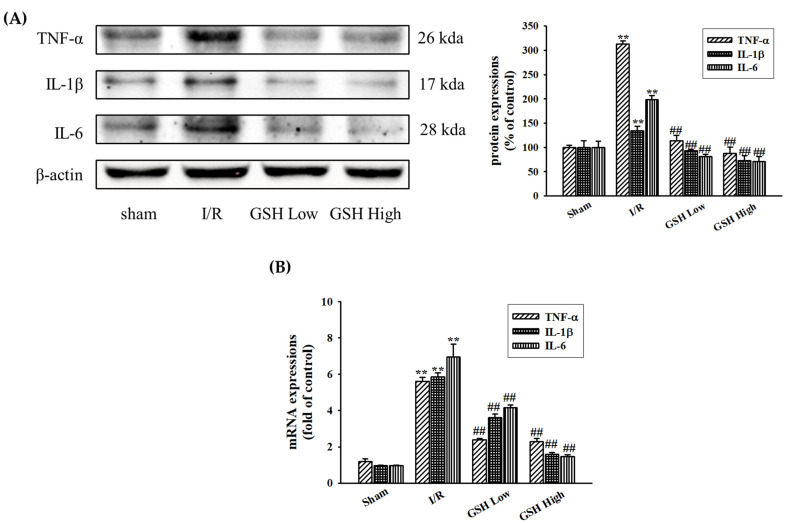
Effects of GSH on proinflammatory cytokines’ expressions in kidney. Protein expressions of cytokines such as TNF-α, IL-1β, and IL-6 were determined by Western blot analysis (**A**). Relative mRNA expression was calculated by real-time RT-qPCR (**B**). Data are presented as means ± SE (*n* = 3). ^##^ *p* < 0.01 vs. sham (normal) group; ** *p* < 0.01 vs. I/R group.

**Figure 6 nutrients-13-03859-f006:**
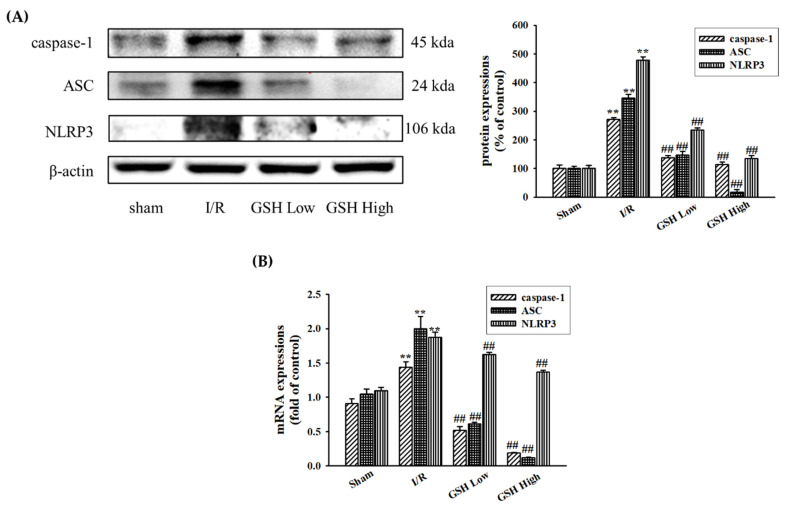
Effects of GSH on NLRP3 inflammasome in kidney. Protein expressions of inflammasome such as caspase-1, ASC, and NLRP3 were analyzed by Western blot analysis (**A**). Relative mRNA expression was analyzed by real-time RT-qPCR (**B**). Data are presented as means ± SE (*n* = 3). ^##^ *p* < 0.01 vs. sham group; ** *p* < 0.01 vs. I/R group.

**Table 1 nutrients-13-03859-t001:** Decoction of Geumgwe-sinkihwan (GSH).

Herbal Name (Part)	Amount (g)	Origin
*Rehmannia glutinosa* (rhizome) 熟地黃	320	Imsil, Korea
*Discorea batatas* (rhizome) 山藥	160	Imsil, Korea
*Cornus officinalis* (fruit) 山茱萸	160	Imsil, Korea
*Poria cocos* (Schw.) *Wolf* (Hoelen) 白茯笭	120	Imsil, Korea
*Alisma orientale* (rhizome) 澤瀉	120	Imsil, Korea
*Paeomia suffructicosa* (cortex) 牧丹皮	120	Imsil, Korea
*Achyranthes aspera* (ro桔梗ot) 牛膝	120	Imsil, Korea
*Plantago ovata* (root) 桔梗	120	Imsil, Korea
Total	1240	

**Table 2 nutrients-13-03859-t002:** Effect of GSH on renal functional parameters.

Group	BUN (mg/dL)	Cr (mg/dL)
sham	29.2 ± 0.55	0.22 ± 0.01
I/R	51.96 ± 7.43 ^#^	0.40 ± 0.04 ^##^
GSH Low	36.46 ± 3.62 **	0.32 ± 0.02 **
GSH High	28.56 ± 0.61 **	0.34 ± 0.01 **

^#^ *p* < 0.05, ^##^ *p* < 0.01 vs. sham group; ** *p* < 0.01 vs. I/R group.

## Data Availability

The data used to support the findings of this study are included within the article.

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
