# Peer review of "Effect of Geumgwe-Sinkihwan on Renal Dysfunction in Ischemia/Reperfusion-Induced Acute Renal Failure Mice"

_nutrients, 2021, doi:10.3390/nu13113859_

Round 1
Reviewer 1 Report
It's always a complex topic to apply Asian traditional medicine to modern disease/basic science. In the present study, the authors tried to show the protective effects of a conventional drug in animal models. It's good to show the culture of one area to the entire world. However, massive critical concerns need to be stated and clarified.
- In the Introduction section, to the readers who are not familiar or don't have traditional medicine background, it's necessary to explain some basic knowledge.
- Since the extract is a mixture of various materials, the localization from the traditional medicine codex (bangyakhappyeon?) should be explained in the method section. For example, is the combination 8:4:4:3:3:3:3:3 in the codex?
- To know the ingredients of this mixture, I checked the name of this drug and found the ingredients vary in Chinese/Japanese/Korean medicine, although the mixture's name is the same.
- It should show the evidence for assuming I/R mice as the type of renal dysfunction in traditional medicine.
- The urine data is critical, but the authors only showed minimal data here. At least, it's requested to show urine volume and Ucr.
- I think in traditional medicine, there should be some indicator from the traditional diagnostic method.
- The histological data is so smeared that it is hard to tell the changes in renal histology. Besides, there was no quantification.
- It's not consistent in Fig 4 showing KIM1 and NGAL expressions in mRNA or protein. It's required to explain or provide the evidence.
- KIM1 was significantly decreased in the medullary section but not the cortex. So is the GSH focused on the medulla?
- All the representative WB bands need to show the MW on the right.
- Fig 7 showed NLRP3 is not acceptable due to the high background and smeared bands. Besides, the ASC in GSH High group was almost nothing but the bar graph showed similar to controlled group.
Author Response
We thank very much to reviewer for the invaluable comments and suggestions. We hope our revision has improved the paper to a level of your satisfaction. We revised the text as red colored.
Reviewer 1
It's always a complex topic to apply Asian traditional medicine to modern disease/basic science. In the present study, the authors tried to show the protective effects of a conventional drug in animal models. It's good to show the culture of one area to the entire world. However, massive critical concerns need to be stated and clarified.
- In the Introduction section, to the readers who are not familiar or don't have traditional medicine background, it's necessary to explain some basic knowledge.
- We appreciate your reviewing our manuscript. The basic contents of traditional medicine treatment using natural products and the objective for this research were added.in the last paragraph of introduction section.
- Since the extract is a mixture of various materials, the localization from the traditional medicine codex (bangyakhappyeon?) should be explained in the method section. For example, is the combination 8:4:4:3:3:3:3:3 in the codex?
- We appreciate your comment. We added the information about decoction of GSH in Bangyakhappyeon in more detail.
“GSH is composed of 8 dried natural herbs; Rehmannia glutinosa (Rhizoma), Discorea batatas (Rhizoma), Cornus officinalis (Fruit), Poria cocos (Schw.) Wolf (Hoelen), Alisma orientale (Rhizoma), Paeomia suffructicosa (cortex), Achyranthes aspera (Radix), Plantago ovata (Radix); in a ratio of 8:4:4:3:3:3:1:1.”
- To know the ingredients of this mixture, I checked the name of this drug and found the ingredients vary in Chinese/Japanese/Korean medicine, although the mixture's name is the same.
- We appreciate your comment. We added the herbal name as Chinese in table 1.
- It should show the evidence for assuming I/R mice as the type of renal dysfunction in traditional medicine.
- We appreciate your comment. There are many previous studies using I/R mice to confirm the efficacy of traditional medicines and natural herbal extracts.
“Preventive Effects of Grape Extract on Ischemia/Reperfusion-Induced Acute Kidney Injury in Mice. M.Ohkita et al. 2019. Biological and Pharmaceutical Bulletin.”
“Renal-Protective Effects and Potential Mechanisms of Traditional Chinese Medicine after Ischemia-Reperfusion Injury. D.Liu et al. 2021. Evidence-based Complementary and Alternative Medicine.”
“Xiaoyu Xiezhuo Drink Protects against Ischemia-Reperfusion Acute Kidney Injury in Aged Mice through Inhibiting the TGF-β1/Smad3 and HIF1 Signaling Pathways. Q. Ye et al. 2021. BioMed Research International.”
“Renal Protective Effect of Beluga Lentil Pretreatment for Ischemia-Reperfusion Injury. S. O. Lee et al. 2021. BioMed Research International.”
- The urine data is critical, but the authors only showed minimal data here. At least, it's requested to show urine volume and Ucr.
- We appreciate for your reviewing. As follow your comment, we added the data about urine volume and UCr
- I think in traditional medicine, there should be some indicator from the traditional diagnostic method.
- We appreciate your comment. In traditional medicine, there are various diagnostic basis such as fever, pain, and etc. However, the relation between these diagnostic basis and diseases has not yet been clearly proven. Therefore, more study is needed on this part.
- The histological data is so smeared that it is hard to tell the changes in renal histology. Besides, there was no quantification.
- We appreciate for your reviewing our manuscript. As follow your comment, we corrected and changed images to more clearly determine glomerular damages and added bar graph.
- It's not consistent in Fig 4 showing KIM1 and NGAL expressions in mRNA or protein. It's required to explain or provide the evidence.
- We appreciate your comment. We re-conducted the experiment and modified the mRNA expression graph.
- KIM1 was significantly decreased in the medullary section but not the cortex. So is the GSH focused on the medulla?
- We appreciate your comment. As your reviewing, the effect in the medullar was greater than in the cortex. However, there were significant decrease in both sites treated with high concentration of GSH. Therefore, GSH can be regarded to be effective by focusing on both medulla and cortex.
- All the representative WB bands need to show the MW on the right.
- We appreciate your reviewing. We added the MW according to the reviewer’s commnet.
- Fig 7 showed NLRP3 is not acceptable due to the high background and smeared bands. Besides, the ASC in GSH High group was almost nothing but the bar graph showed similar to controlled group.
- We appreciate your comment. We modified the band image quality more clearly and the bar graph was corrected by re-quantification.
Reviewer 2 Report
I would like to know about the method of feeding animals. Did they have 24/7 access to the tested mixture of herbs, or were the herbs administered through some kind of gastric tube? On what basis was it known how much they consumed the mixtures of these herbs (water solution as I understand it). How was it monitored? I understand that the appropriate doses specified by You are marked as GSH low, i.e. 100mg / kg / day and GSH high, i.e. 300mg / kg / day - then kg of mouse body weight, right? Therefore, whether the mice were weighed every time - because it is known that there is an increase in weight. Were the mice kept in individual or group cages?
Author Response
We thank very much to reviewer for the invaluable comments and suggestions. We hope our revision has improved the paper to a level of your satisfaction. We revised the text as red colored.
Reviewer 2
I would like to know about the method of feeding animals. Did they have 24/7 access to the tested mixture of herbs, or were the herbs administered through some kind of gastric tube?
- We appreciate your comment. GSH was treated with drinking water to mice.
On what basis was it known how much they consumed the mixtures of these herbs (water solution as I understand it). How was it monitored? I understand that the appropriate doses specified by You are marked as GSH low, i.e. 100mg / kg / day and GSH high, i.e. 300mg / kg / day - then kg of mouse body weight, right?
- The amount of water consumed by mice every day during acclimatization was measured. Based on the results obtained by calculating the amount of water (water intake – water out) and the weight of the mice, GSH was provided to the mice. In addition, as you mentioned, Kg means mice body weight.
Therefore, whether the mice were weighed every time - because it is known that there is an increase in weight.
- The weight of the animal was measured before surgery. And GSH was provided for 3 days based on its weight.
Were the mice kept in individual or group cages?
- We appreciate your comment. All mice were divided into groups and housed separately in a metabolic cage. This is mentioned in Materials and Methods section (2.2).